# Gene Expression Profiling Reveals that PXR Activation Inhibits Hepatic PPARα Activity and Decreases FGF21 Secretion in Male C57Bl6/J Mice

**DOI:** 10.3390/ijms20153767

**Published:** 2019-08-01

**Authors:** Sharon Ann Barretto, Frédéric Lasserre, Anne Fougerat, Lorraine Smith, Tiffany Fougeray, Céline Lukowicz, Arnaud Polizzi, Sarra Smati, Marion Régnier, Claire Naylies, Colette Bétoulières, Yannick Lippi, Hervé Guillou, Nicolas Loiseau, Laurence Gamet-Payrastre, Laila Mselli-Lakhal, Sandrine Ellero-Simatos

**Affiliations:** Institut National de la Recherche Agronomique (INRA), UMR1331 Toxalim, F31-027 Toulouse CEDEX 3, France

**Keywords:** nuclear receptors, hepatokines, transcriptomics

## Abstract

The pregnane X receptor (PXR) is the main nuclear receptor regulating the expression of xenobiotic-metabolizing enzymes and is highly expressed in the liver and intestine. Recent studies have highlighted its additional role in lipid homeostasis, however, the mechanisms of these regulations are not fully elucidated. We investigated the transcriptomic signature of PXR activation in the liver of adult wild-type vs. *Pxr*^-/-^ C57Bl6/J male mice treated with the rodent specific ligand pregnenolone 16α-carbonitrile (PCN). PXR activation increased liver triglyceride accumulation and significantly regulated the expression of 1215 genes, mostly xenobiotic-metabolizing enzymes. Among the down-regulated genes, we identified a strong peroxisome proliferator-activated receptor α (PPARα) signature. Comparison of this signature with a list of fasting-induced PPARα target genes confirmed that PXR activation decreased the expression of more than 25 PPARα target genes, among which was the hepatokine fibroblast growth factor 21 (*Fgf21*). PXR activation abolished plasmatic levels of FGF21. We provide a comprehensive signature of PXR activation in the liver and identify new PXR target genes that might be involved in the steatogenic effect of PXR. Moreover, we show that PXR activation down-regulates hepatic PPARα activity and FGF21 circulation, which could participate in the pleiotropic role of PXR in energy homeostasis.

## 1. Introduction

Pregnane X receptor (PXR, systematic name NR1I2) is a member of the nuclear receptor superfamily and is highly expressed in the liver and intestine of mammals [1]. PXR was characterized as a xenosensor that regulates the expression of xenobiotic-metabolizing enzymes and transporters, thereby facilitating the elimination of xenobiotics and endogenous toxic chemicals such as bile acids [2]. Upon ligand-binding, PXR translocates to the nucleus, heterodimerizes with retinoid X receptor (RXR, NR2B1) and binds to PXR direct repeat 4 (DR-4) response elements (PXRE) that are usually located upstream of target genes. Because of an unusually large and flexible binding pocket, PXR can be activated by a variety of structurally diverse chemicals, including pharmaceutical drugs, dietary supplements, herbal medicines, environmental pollutants, and endogenous molecules [3]. In line with the role of PXR as a master regulator of xenobiotic metabolism, its first described target gene was cytochrome P450 (CYP) 3A4 in humans [4], which represents 10% of all clinically relevant drug-metabolizing CYPs in the human liver and up to 75%–85% in the intestine [5] and is responsible for the metabolization of 60% of marketed drugs [6].

Besides its original function as part of the detoxification machinery, recent studies have also unveiled functions for PXR in intermediary metabolism. There is an increasing amount of clinical evidence showing that PXR agonists cause hyperglycemia in humans [7] and pre-clinical work suggesting that PXR regulates hepatic glucose metabolism, however, there is still no solid understanding of the consequences, or of the mechanisms involved. Activated PXR has been shown to repress expression of the gluconeogenic genes glucose-6-phosphatase (*G6Pase*) and phosphoenolpyruvate carboxykinase (*PCK1*) [8], and of genes involved in glucose uptake such as *GLUT2* and of glucokinase (*GCK*) [9]. Although there is limited data on the relationship between PXR and fatty liver in humans in vivo, many studies have demonstrated that PXR activation also causes hepatic lipid accumulation in human cell models, and in vitro and in vivo mouse models [7,10]. This pro-steatotic effect is thought to result from both the activation of lipogenesis and inhibition of ß-oxidation [7]. However, the mechanisms by which PXR activation induces perturbations of lipid metabolism are not fully elucidated. Recently, it was shown that the activation of intestinal PXR signaling induced dyslipidemia and intestinal cholesterol accumulation [11], while activation of hepatic PXR signaling was sufficient to promote hypercholesterolemia and hepatic lipid accumulation [12].

Here, we aimed to gain insights into the potential metabolic dysregulations induced upon PXR activation and performed a transcriptomic comparison of the hepatic gene profiles of wild type (WT) vs. *Pxr*^-/-^ male mice treated with the rodent specific PXR ligand pregnenolone 16α-carbonitrile (PCN). As expected, we observed that PCN treatment-induced hepatic steatosis. We unraveled several previously unknown PXR target genes involved in liver lipid accumulation and discovered a very robust peroxisome proliferator-activated receptor α (PPARα) signature amongst the PXR down-regulated target genes. The PXR-induced decrease in PPARα activity included the regulation of the hepatokine FGF21, a liver-derived hormone with major endocrine roles [13]. This cross-talk between PXR and PPARα in the regulation of FGF21 may contribute to endocrine disruption by xenobiotics acting as ligands for PXR.

## 2. Results

### 2.1. Effect of PXR Activation on Physiological Parameters and Liver Lipids

We investigated the effect of PXR activation by its pharmacological ligand PCN in WT and *Pxr*^-/-^ male mice. PCN treatment did not affect body weight but increased relative liver mass in a PXR-dependent way (Figure 1a). In the liver, PXR activation significantly increased cholesterol esters and triglyceride levels but did not significantly impact free cholesterol (Figure 1a). In the plasma, PXR activation increased alanine transaminase (ALT) and decreased total cholesterol levels but did not impact free fatty acids, triglycerides (Figure 1a), HDL, LDL, or glucose levels (Appendix A).

### 2.2. Effects of PXR Activation on the Hepatic Transcriptome

Using microarrays, we obtained global transcriptional profiles. Principal component analysis (PCA) first illustrated that PCN treatment significantly impacted the hepatic transcriptome (Figure 2a). The discrimination of WT PCN vs. WT Cont seems stronger than that of the *Pxr*^-/-^ PCN vs. *Pxr*^-/-^ Cont, confirming, as expected, a significant PXR-dependent transcriptional effect of PCN. We next used linear models and considered genes to be significantly regulated with a fold-change >1.5 and a false discovery rate (FDR) <0.05. Heatmap clustering confirmed the PCA results (Appendix A). It indeed revealed five gene clusters with the largest cluster (1602 probes) comprised of genes up-regulated by PCN in WT mice only (cluster 5). Another cluster (cluster 2) showed 407 probes down-regulated upon PCN treatment in WT mice only. Cluster 4 contained 498 probes that showed genes differentially regulated in WT vs. *Pxr*^-/-^ mice, independently of PCN. Finally, cluster 3 (605 probes) illustrated a PCN effect in both WT and *Pxr*^-/-^ mice.

We next sought to decipher the biological functions affected by PXR activation. PCN treatment significantly up-regulated the expression of 1258 genes in WT animals, and of 333 genes in *Pxr*^-/-^ mice (Figure 2c). Using the 1029 “prototypical” PXR target genes (those that were up-regulated only in WT animals), we conducted a pathway enrichment analysis, which revealed seven functional clusters significantly enriched (Figure 2d and Appendix A) with genes involved in cell cycle, cell division and mitosis, glutathione metabolism, cytochromes P450, lipid metabolism, chemotaxis, and positive regulation of inflammatory response. Figure 2e confirms these results by illustrating the fold-changes of the top 30 most up-regulated genes. These results first confirmed the well-described influence of PXR activation on hepatic xenobiotic-metabolizing enzymes, mainly those from the Cyp3 family. Appendix A provides a full description of the impact of PCN treatment on all xenobiotic-metabolizing enzymes. Induction of two of the most well-described PXR targets, *Cyp2c55* and *Cyp3a11* were further confirmed using RT-qPCR (Figure 2b). Interestingly, the “lipid metabolism” pathway was also highly significantly enriched upon PXR activation and, among the 30 genes with the highest fold-change, the patatin-like phospholipase domain containing 3 (*Pnpla3*), the thyroid hormone-responsive spot 14 (*Thrsp* or *Spot14*), and the growth/differentiation factor 15 (*Gdf15*) belonged to this pathway. Induction of *Gdf15* was also confirmed by RT-qPCR (Figure 2b). Finally, the regulation of genes involved in de novo lipogenesis was also confirmed by qPCR and showed a significant increase of the SREBP-1 lipogenic pathway in *Pxr*^-/-^ mice compared to WT mice (Appendix A).

We next investigated the effect of PCN on gene down-regulation. PCN treatment significantly decreased the expression of 186 genes in a PXR-dependent manner (Figure 2f). GO analyses revealed that these genes were involved in lipid metabolic process, biological rhythms, transforming growth factor-β (TGFβ) signaling pathway, glucose metabolism, and cytochromes P450 (Figure 2g). The 30 genes with the highest fold-changes are illustrated in Figure 2h. Interestingly, among these 30 genes, five (namely *Fgf21*, *Cyp4a10*, *Cyp4a31*, *Acot1,* and *Plin4*) are well-described target genes of PPARα, a key hepatic transcriptional regulator involved in lipid homeostasis.

### 2.3. Comparison of PXR and PPARα-Dependent Transcriptome

This prompted us to investigate the intersection between PXR and PPARα activation to test the hypothesis that PXR activation influenced PPARα activity. We took advantage of our previously published microarray dataset [14], in which C57Bl6/J male mice carrying an hepatocyte-specific deletion of *Pparα* (*Pparα*^hep-/-^) were fasted for 24 h to induce PPARα activity and compared to their wild-type littermates (*Pparα*^hep+/+^). We have indeed previously shown that, during fasting, PPARα senses increased levels of free fatty acids released from adipocytes, and in response, controls the expression of hundreds of genes involved in fatty acid uptake, transport, and catabolism in hepatocytes [14,15]. Figure 3a indeed illustrates that a large number of the fasting-induced hepatic genes are PPARα sensitive, with 538 genes significantly up-regulated in a PPARα-dependent manner. Also, 461 genes were significantly down-regulated in a PPARα-dependent manner upon fasting (Figure 3d). We compared these genes with those regulated upon PXR activation. We found 27 genes that were both up-regulated upon PPARα activation and down-regulated upon PXR activation (Figure 3b). These genes are illustrated in Figure 3c and include, among others, *Pparα* itself, *Cyp4a14*, *Cyp4a10*, *Cyp4a31,* and *Fgf21*. There were also 46 genes that were regulated in the opposite direction, i.e., that were down-regulated upon PPARα activation and up-regulated upon PXR activation (Figure 3e,f).

### 2.4. Regulation of FGF21

Using RT-qPCR analyses, we confirmed that PXR activation down-regulated *Pparα* and its target genes expression (Figure 4a), among which was *Fgf21*. FGF21 is a recently described hepatokine with systemic metabolic effects [16]. We measured plasmatic FGF21 and confirmed that circulating FGF21 was decreased upon PCN treatment, since its levels were not detectable anymore in WT-treated mice (Figure 4b). Surprisingly, PXR deletion also influenced FGF21 level since *Pxr*^-/-^ mice also showed no detectable levels of circulating FGF21. These differences were not due to different fasting states since glycemia was not significantly different between the four groups (Appendix A).

## 3. Discussion

The liver is one of the major organs involved in energy production. Hepatic lipid metabolism plays a crucial role during fasting and/or prolonged exercise. Upon lowering of blood glucose, the liver increases glucose production by augmenting gluconeogenesis and glycogenolysis to maintain blood glucose levels, increases fatty acid oxidation and ketogenesis to provide extra-hepatic tissues with ketone bodies, and decreases lipogenesis to attenuate triglyceride storage. These processes are under tight transcriptional control and, in response to hormones such as glucagon and glucocorticoids, many transcription factors cooperate to regulate various genes involved in metabolic pathways aimed at restoring homeostasis [17]. Among those, hepatic PPARα has been well described as crucial for this adaptation. However, recent data have highlighted that other nuclear receptors, such as the aryl H receptor (AhR), the constitutive androstane receptor (CAR), and PXR, which were historically described as xenobiotic sensors, can also interact with the hormone-responsive transcription factors to regulate the liver metabolic processes [18].

Here, we investigated the transcriptomic effects of a pharmacological activation of PXR. The expression of PXR was not described as highly circadian, however, its activity, as measured by the expression of its prototypical target gene *Cyp3a11*, has been shown to be influenced by the time of the day, and is highest as zeitgeber time (ZT)6 [19]. Therefore, we decided to investigate the effects of PXR activation at ZT6, a time at which mice were in a physiological semi-fasted state.

Several studies have already investigated the hepatic signature of PXR activation in vivo [20,21,22] or in vitro [23]. However, most of these studies focused on the effect of PXR activation on xenobiotic-metabolizing enzymes. Here, we confirm that the regulation of xenobiotic metabolism is one of PXR’s most potent functions in the hepatocytes (Figure 2; Appendix A). However, our gene enrichment analyses also revealed that lipid metabolism was among the top-dysregulated pathways upon PXR activation, considering both the up-regulated, as well as the down-regulated genes.

First, PXR activation induced a very significant decrease in plasma cholesterol levels and a significant increase in liver triglycerides and cholesterol esters (Figure 1). The pro-steatotic effects of acute PXR activation have been shown in many studies. However, its role in the regulation of cholesterol homeostasis is more controversial. The anti-HIV drug Efavirenz has been recently shown to induce steatosis and hypercholesterolemia, an effect that was absent in a model of hepatic deletion of PXR [12]. These perturbations were mediated through increased fatty acid transport and cholesterol synthesis, via the PXR-dependent regulation of *Cd36* and *Sqle*. In our data, we confirmed that PXR activation significantly affected *Cd36* and other transporters involved in cholesterol transport, but did not observe any regulation of genes involved in cholesterol biosynthesis, such as *Cyp7a1*, *Sqle,* and *Hmgcr* (Appendix A). This resulted in decreased circulating cholesterol.

Among the up-regulated genes in the liver, we observed that PXR activation increased the expression of several genes that correlate with lipogenesis, such as the patatin-like phospholipase domain containing 3 (*Pnpla3*) and the thyroid hormone-responsive spot 14 (*Thrsp* or *Spot14*). *Spot14* was first identified as a thyroid-responsive gene and is known to transduce hormone- and nutrient-related signals to genes involved in lipogenesis [24]. Regulation of *SPOT14* by PXR was previously described in human hepatocytes [25] and led to increased fatty acid synthase (FASN) expression and triglyceride accumulation. The PNPLA3 protein has lipase activity towards triglycerides in hepatocytes and a loss-of-function polymorphism of this gene has been shown to be strongly associated with nonalcoholic fatty liver disease [26]. However, to our knowledge, the regulation of *Pnpla3* expression by PXR has not been previously described. Among the lipid-metabolic-related genes, we also observed that the expression of the growth/differentiation factor 15 (*Gdf15*), also known as *MIC-1*, was increased by a factor of four upon PCN treatment, in a PXR-dependent way. GDF15 is a distant member of the transforming growth factor-ß (TGF-ß) superfamily that is considered a crucial hormone in regulating lipid and carbohydrate metabolism. In animal models, overexpression of GDF15 leads to a lean phenotype and improvements of metabolic parameters by increasing the expression of key thermogenic and lipolytic genes in brown and white adipose tissue [27]. Hepatic and circulating GDF15 levels were also increased in animals with blunted ß-oxidation (*Cpt2*^hep-/-^ mice) to maintain systemic energy homeostasis upon fasting [28]. Whether the observed increase in *Gdf15* mRNA upon PCN treatment results from direct regulation of *Gdf15* by PXR or represents a secondary adaptation to decreased ß-oxidation remains to be determined. In both cases, regulation of GDF15 levels upon PXR activation might be of physiological relevance since GDF15 has been implicated in a wide variety of biological functions including control of food intake and body weight [29].

Among the genes that were down-regulated upon PXR activation, we observed a very consistent PPARα-like signature, with the decreased expression of many *Cyp4* genes, which are highly sensitive PPARα target genes [14,15]. These results coincide with previous findings in which PCN decreased the hepatic expression of *Pparα*, *Cyp4a10,* and *Cyp4a14* [21]. Neonatal exposure to a single dose of PCN also persistently down-regulated *Cyp4a* expression and decreased PPARα binding to the *Cyp4a* gene loci in adult mice [20]. By comparing the list of genes down-regulated upon PXR activation to a list of genes up-regulated upon PPARα activation, we here extend these previous findings and demonstrate that the inhibition of PPARα activity by PXR affects more than the expression of *Cyp4* genes. For example, the PXR–PPARα interaction probably inhibited the expression of the acetyl-Coenzyme A acyltransferase 1B (*Acaa1b*), of the acyl-coA thioesterase 3 (*Acot3*), of *Krt23* and *Rab30*, of the rate-limiting enzyme in ketogenesis 3-hydroxy-3-methylglutaryl-CoenzymeA synthase 2 (*Hmgcs2*) and of the hepatokine *Fgf21*, all of which are well-described PPARα targets [14]. Using a similar approach in human primary hepatocytes treated with the hPXR ligand rifampicine and the hPPARα ligand WY14643, Kandel et al. had previously shown that more than 14 genes were responsive to both WY14643 (up-regulated) and to rifampicine (down-regulated), among which ACAA2, CYP4A11, and HMGCS2 [23], therefore suggesting the human relevance of our results.

FGF21 is predominantly produced in the liver [30] and exerts pleiotropic effects on the body to maintain overall metabolic homeostasis. FGF21 metabolic benefits range from reducing body weight to alleviating hyperglycemia, insulin resistance, and improvement of lipid profiles [16]. In animal models of obesity, as well as in obese patients, FGF21 has been shown to induce body weight loss and to increase insulin sensitivity and lipid homeostasis [30]. The effects of FGF21 on fertility, growth, and longevity are also well documented [31,32]. Finally, FGF21 seems to be involved in food preferences. For example, FGF21 production in response to carbohydrate intake significantly decreases sugar preferences [33].

Although PXR is mainly expressed in the liver and in the intestine, and not in adipose tissue [34], deletion of *Pxr* appears to influence insulin sensitivity in white adipose tissue and in the muscle [35], serum leptin, and adiponectin levels [36] and PXR activation regulates gene expression in both white and brown adipose tissues [37]. This suggests systemic effects of *Pxr* deletion and activation for which mechanisms have not been described yet. White and brown adipose tissues are among the most described target tissues of FGF21 [16]. Whether FGF21 could be an effector of the systemic effects of PXR remains an open question. Here, we demonstrate that both PXR-activation and PXR deletion decrease the hepatic *Fgf21* mRNA levels and completely abolished the circulating FGF21 levels. This apparent contradictory effect was not limited to the regulation of FGF21 but was also observed in other PPARα target genes (Figure 4). Therefore, it seems that both PXR activation and silencing result in the inhibition of PPARα activity, probably through distinct mechanisms that would need additional investigations. However, it is worth noticing that the same apparent contradictory effect was observed for the regulation of de novo lipogenesis. In human HepG2 cells, PXR activation by rifampicin promoted steatosis via induction of SREBP-1 pathway (mainly SREBP-1a), whereas PXR silencing enhanced AKR1B10 expression, which subsequently stabilized the acetyl-CoA carboxylase, thereby promoting de novo lipogenesis [10]. However, these mechanisms are probably species-specific as, in our data, we did not observe this increase in AKR1B10 expression, whereas the SREBP-1 pathway was increased by PXR ablation and not by PCN treatment (Appendix A). Overall, this demonstrated that complex species-specific mechanisms occur in the regulation of lipogenic pathways by PXR activation and ablation, and our results suggest that this might also be true for the regulation of ß-oxidation and PPARα activity.

Perspectives and limitations of our study include the use of male mice only, while PXR activation has been shown to impact both xenobiotic-metabolizing enzymes and glucose and lipid metabolism in a sexually-dimorphic way [38,39]. Therefore, it would be interesting to decipher whether the signature of PXR activation described in our study is also valid in female mice. Second, our study focused on short-term changes. An important remaining question is to determine the effect of multiple weak PXR agonists such as those present in our environment on the observed regulations, especially on FGF21 secretion. Indeed, PXR’s main target gene *CYP3A4* is known to be involved in the metabolism of more than 60% of the currently marketed drugs [6] and several hundreds of environmental, occupational, and natural products are demonstrated PXR agonists in both mice and humans [3]. Therefore, regulation of hepatic lipid accumulation by acute or chronic PXR activation might be an important mechanism of xenobiotic-induced steatosis. Finally, the fact that we did not generate the PXR and PPARα dependent transcriptomes in a parallel fashion might have underestimated the number of genes affected by the cross-talk between the two receptors. Therefore, it would be interesting to investigate the effect of PXR activation upon prolonged fasting such as the one used to trigger PPARα. It could also be interesting to decipher whether the pro-steatotic effect of PCN depends on PPARα by treating PPARα knock-out mice with PCN.

Altogether, our results present an additional resource of transcriptome analyses that confirm and extend previous findings on the genes involved in the pro-steatotic effects of PXR. As previously observed in various models [7], we confirm that the observed pro-steatotic effect of PXR activation probably results from both induction of lipogenesis and repression of β-oxidation, and further highlight that this repression is certainly mediated, at least in part, through inhibition of PPARα. We also provide new hypotheses regarding the yet poorly explored pleiotropic effects of PXR that could result from the regulation of recently discovered hepatokines, such as GDF15 and/or FGF21. More studies are needed to confirm the physiological relevance of these regulations. Our findings might have clinical and public health relevance given the wide range of drugs and environmental xenobiotics that have been described as PXR ligands and potential endocrine disruptors.

## 4. Materials and Methods

### 4.1. Animals

In vivo studies were performed in a conventional laboratory animal room following the European Union guidelines for laboratory animal use and care. The current project was approved by an independent ethics committee (CEEA-86 Toxcométhique) under the authorization number 2018062810452910. The animals were treated humanely with due consideration to the alleviation of distress and discomfort. All mice were housed at 21–23 °C on a 12 h light (ZT0–ZT12) 12 h dark (ZT12–ZT24) cycle and allowed free access to the diet (Teklad Global 18% Protein Rodent Diet) and tap water. ZT stands for Zeitgeber time; ZT0 is defined as the time when the lights are turned on. Twelve six-week-old wild-type (WT) C57BL/6J male mice were purchased from Charles River and 12 *Pxr*^-/-^ animals (backcrossed on the C57Bl/6J background) were engineered in Pr. Meyer’s laboratory [40] and were bred for 10 y in our animal facility. Mice were acclimatized for two weeks, then randomly allocated to the different experimental groups: Wild-type control (WT CONT, *n* = 6), wild-type PCN-treated (WT PCN, *n* = 6), *Pxr*^-/-^ control (*Pxr*^-/-^ CONT, *n* = 6), *Pxr*^-/-^ PCN-treated (*Pxr*^-/-^ PCN, *n* = 6). PCN-treated mice received a daily intraperitoneal injection of PCN (100 mg/kg) in corn oil for 4 days while control mice received corn oil only. Mice were killed at ZT6, 6 h after the last PCN injection.

### 4.2. Blood and Tissue Samples

Bodyweight was monitored at the beginning and at the end of the experimental period. Prior to sacrifice, the submandibular vein was lanced, and blood was collected into lithium heparin-coated tubes (BD Microtainer, Franklin Lake, NJ, USA). Plasma was prepared by centrifugation (1500 g, 10 min, 4 °C) and stored at −80 °C. At sacrifice, the liver was removed and snap-frozen in liquid nitrogen and stored at −80 °C until used for RNA extraction.

### 4.3. Gene Expression

Total RNA was extracted with TRIzol reagent (Invitrogen, Carlsbad, CA, USA). Gene expression profiles were obtained at the GeT-TRiX facility (GénoToul, Génopole Toulouse Midi-Pyrénées, France) using Sureprint G3 Mouse GE v2 microarrays (8 × 60 K; design 074,809; Agilent Technologies, Santa Clara, CA, USA) following the manufacturer’s instructions Microarray data and experimental details are available in NCBI’s Gene Expression Omnibus [41] and are accessible through GEO Series accession numbers GSE123804. For real-time quantitative polymerase chain reaction (qPCR), 2 μg RNA samples were reverse-transcribed using the High-Capacity cDNA Reverse Transcription Kit (Applied Biosystems, Foster City, CA, USA). Appendix A presents the SYBR Green assay primers. Amplifications were performed using an ABI Prism 7300 Real-Time PCR System (Applied Biosystems, Foster City, CA, USA). qPCR data were normalized to TATA-box-binding protein mRNA levels and analyzed with LinRegPCR.v2015.3.

### 4.4. Plasma Analysis

Alanine transaminase (ALT), total cholesterol, triglycerides and free fatty acids (FFA) were determined using a Pentra 400 biochemical analyzer (Anexplo facility, Toulouse, France). Plasma FGF21 was assayed using the rat/mouse FGF21 ELISA kit (EMD Millipore, Billerica, MA, USA) following the manufacturer’s instructions.

### 4.5. Liver Neutral Lipid Analysis

Tissue samples were homogenized in methanol/5 mM EGTA (2:1, *v/v*); then, lipids (corresponding to an equivalent of 2 mg tissue) were extracted according to the Bligh and Dyer method [42] with chloroform/methanol/water (2.5:2.5:2.1, *v/v/v*), in the presence of the following internal standards: glyceryl trinonadecanoate, stigmasterol, and cholesteryl heptadecanoate (Sigma, Saint-Louis, MO, USA). Triglycerides, free cholesterol, and cholesterol esters were analyzed with gas–liquid chromatography on a Focus Thermo Electron system equipped with a Zebron-1 Phenomenex fused- silica capillary column (5 m, 0.25 mm i.d., 0.25 mm film thickness). The oven temperature was programmed to increase from 200 to 350 °C at 5 °C/min, and the carrier gas was hydrogen (0.5 bar). Injector and detector temperatures were 315 °C and 345 °C respectively.

### 4.6. Statistical Analysis

Microarray data were processed using R (http://www.r-project.org, accessed at 22 September 2017) and Bioconductor packages (www.bioconductor.org, accessed at 22 September 2017, v 3.0). Raw data (median signal intensity) were filtered, log2 transformed, corrected for batch effects (microarray washing bath), and normalized using CrossNorm method [43]. Normalized data were first analyzed using Matlab (v2014.8). The principal component analysis was performed using an in-house function. The linear model was fitted using the limma lmFit function [44]. Pair-wise comparisons between biological conditions were applied using specific contrasts. A correction for multiple testing was applied using the Benjamini–Hochberg procedure for false discovery rate (FDR). Probes with FDR ≤ 0.05 and |fold-change| > 1.5 were considered to be differentially expressed between conditions. Gene-annotation enrichment analysis and functional annotation clustering were evaluated using DAVID [45]. For non-microarray data, differential effects were analyzed by analysis of variance followed by Tukey’s post-hoc tests. A *p*-value < 0.05 was considered significant.

## Figures and Tables

**Figure 1 ijms-20-03767-f001:**
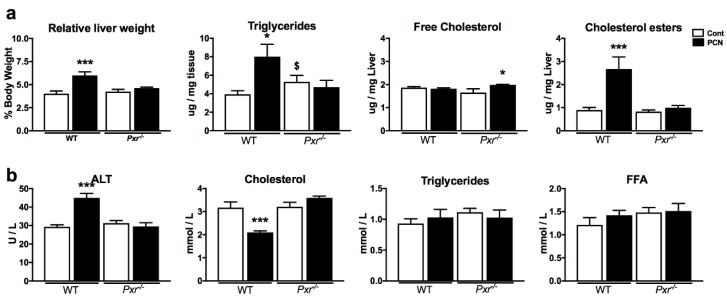
Effect of pregnenolone 16α-carbonitrile (PCN) treatment on liver parameters (**a**) and plasma biochemistry (**b**). Data are shown as mean ± SEM of *n* = 5–6 per group. * *p* ≤ 0.05, ** *p* ≤ 0.01, *** *p* ≤ 0.005 for PCN effect using 2-way ANOVA and Tukey’s post-tests. $ *p* ≤ 0.05 for genotype effect. ALT: Alanine amino-transferase; FFA: Free fatty acids.

**Figure 2 ijms-20-03767-f002:**
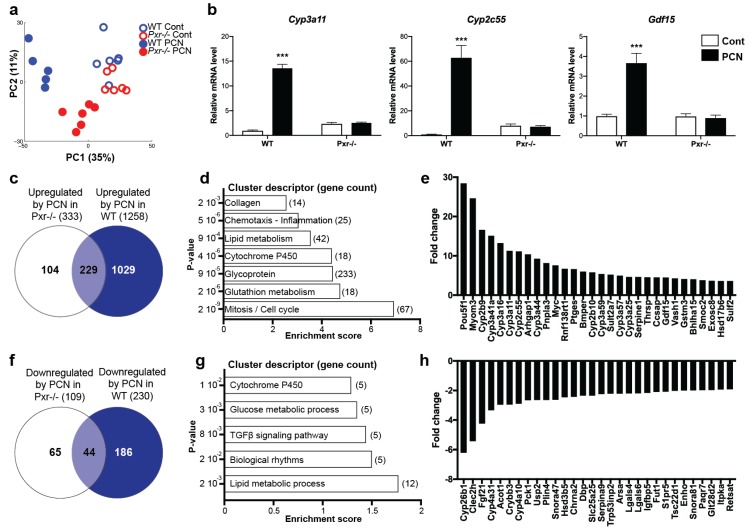
Impact of pregnane X receptor (PXR) activation on the hepatic transcriptome. (**a**) Principal component analysis (PCA) score plots of the whole transcriptomic dataset. (**b**) qPCR confirmation on selected genes. * *p* ≤ 0.05, ** *p* ≤ 0.01, *** *p* ≤ 0.005 for PCN effect using 2-way ANOVA and Tukey’s post-tests. (**c**,**f**) Venn diagram representing the number of genes affected by PCN treatment. (**d**,**g**) Gene enrichment analyses of the PXR-target genes. (**e**,**h**) The 30 genes with the highest fold-changes upon PCN treatment.

**Figure 3 ijms-20-03767-f003:**
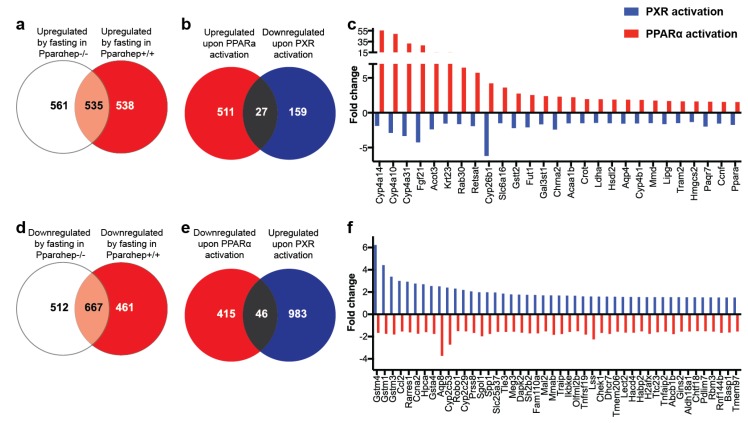
Comparison between PXR and peroxisome proliferator-activated receptor α (PPARα) target genes. (**a**,**d**) Venn diagrams representing the number of genes up-(**a**) or down-(**d**) regulated upon fasting in Pparα^hep+/+^ vs. Pparα^hep-/-^ mice. (**b**,**e**) Venn diagrams representing the number of genes regulated upon PPARα (red) or PXR (blue) activation. (**c**,**f**) Fold-changes for the genes that are shared in the previous Venn diagrams.

**Figure 4 ijms-20-03767-f004:**
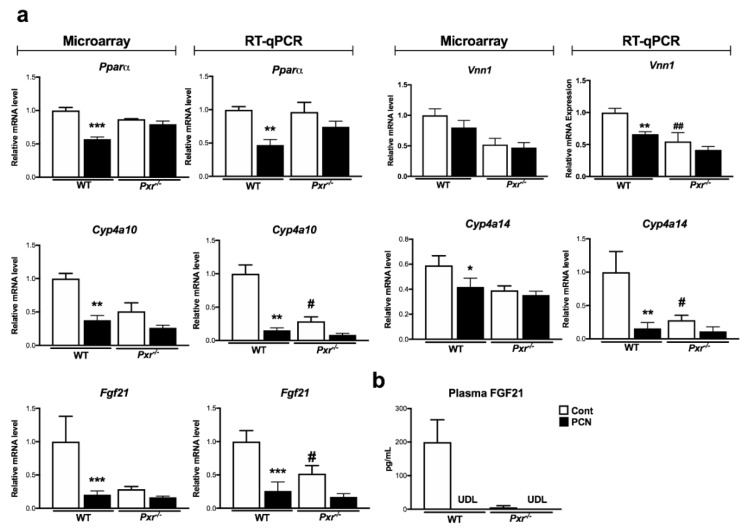
Impact of PXR activation on hepatic PPARα activity. Gene expression in the liver (**a**) derived from the microarray and from complementary qPCR experiments. (**b**) Plasma levels of FGF21. Data are mean ± SEM of *n* = 5–6 per group. * *p* ≤ 0.05, ** *p* ≤ 0.01, *** *p* ≤ 0.005 for PCN effect, # *p* ≤ 0.05, ## *p* ≤ 0.01 for genotype effect using 2-way ANOVA and Tukey’s post-tests. UDL: Under the detection limit.

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
