# Peer review of "Gene Expression Profiling Reveals that PXR Activation Inhibits Hepatic PPARα Activity and Decreases FGF21 Secretion in Male C57Bl6/J Mice"

_ijms, 2019, doi:10.3390/ijms20153767_

Round 1
Reviewer 1 Report
Activation of PXR induces lipogenesis, inhibits fatty acid β-oxidation and inhibits gluconeogenesis. FGF21 is a peptide hormone that is synthesized by several organs and regulates energy homeostasis. In this study, they showed PXR activation inhibits hepatic PPARα activity and decreases FGF21 secretion in male mice. There are several issues the authors need to address.
1. In 2.1 section, the author should detect liver function (ALT and AST).
2. In figure 1B, cholesterol and free fatty acids (FFA) are different kinds of lipid molecules. If they could detect cholesterol, free cholesterol, free fatty acids, it will be better.
3. Some known PXR target gene associated with energy homeostasis should shown in their microarray and qPCR results, such as lipin-1, SCD-1, CD36, SREBP-1…
4. For the microarray data, they should use heatmap to show more changed data.
5. In 2.3 section, how to activation PXR in Pparα (Pparαhep-/-) mice, is it treated with PCN in Pparαhep-/- mice? I'm confused about this part.
6. FGF21 is a hepatokine – i.e., a hormone secreted by the liver. The author should detect FGF21 protein level in liver. And the author should consider that fasting time may influence FGF21 Level.
Author Response
We would like to thank the reviewer for his/her insightful comments. We believe that his/her suggestions have increased the scientific quality and clarity of this manuscript.
Reviewer 1
Activation of PXR induces lipogenesis, inhibits fatty acid β-oxidation and inhibits gluconeogenesis. FGF21 is a peptide hormone that is synthesized by several organs and regulates energy homeostasis. In this study, they showed PXR activation inhibits hepatic PPARα activity and decreases FGF21 secretion in male mice. There are several issues the authors need to address.
1. In 2.1 section, the author should detect liver function (ALT and AST).
Plasma ALT are provided in Figure 1b. We have conducted additional plasma measurements and now show plasma AST levels in Supplementary Figure 1. Plasma AST show the same trend as plasma ALT.
2. In figure 1B, cholesterol and free fatty acids (FFA) are different kinds of lipid molecules. If they could detect cholesterol, free cholesterol, free fatty acids, it will be better.
We have conducted additional plasma measurements and have added plasmatic FFA levels in Figure 1b, and HDL and LDL levels in Supplementary Figure 1.
We have also measured total hepatic cholesterol and included these results in Figure 1a.
The description of these additional parameters was also added in the corresponding result paragraph (lines 89-96).
3. Some known PXR target gene associated with energy homeostasis should shown in their microarray and qPCR results, such as lipin-1, SCD-1, CD36, SREBP-1…
We thank the reviewer for this comment. We have now added two Supplementary Figures with the microarray results and confirmatory qPCR experiments for genes involved in cholesterol transport and synthesis (Supplementary Figure 4) and for genes involved in de novo lipogenesis (Supplementary Figure 3).
We also commented these figures in the result section (lines 141-144): “Finally, regulation of genes involved in de novo lipogenesis was also confirmed by qPCR and showed a significant increase of the SREBP-1 lipogenic pathway in Pxr-/- mice compared to WT mice (Supplementary Figure 3)” and in the discussion section (lines 268-280): “First, PXR activation induced a very significant decrease in plasma cholesterol levels, and a significant increase in liver triglycerides and cholesterol esters (Figure 1). The pro-steatotic effects of acute PXR activation have been shown in many studies. However, its role in the regulation of cholesterol homeostasis is more controversial. The anti-HIV drug Efavirenz has been recently shown to induce steatosis and hypercholesterolemia, an effect that was absent in a model of hepatic deletion of PXR (12). These perturbations were mediated through increased fatty acid transport and cholesterol synthesis, via the PXR-dependent regulation of Cd36 and Sqle. In our data, we confirmed that PXR activation significantly affected Cd36 and other transporters involved in cholesterol transport, but did not observe any regulation of genes involved in cholesterol biosynthesis such as Cyp7a1, Sqle and Hmgcr (Supplementary Figure 4). This resulted in decreased circulating cholesterol.”
4. For the microarray data, they should use heatmap to show more changed data.
We thank the reviewer for this suggestion and added a heatmap of our microarray data in Supplementary Figure 2. The following description was also included in the text (lines 115-121): “Heatmap clustering confirmed the PCA results (Supplementary Figure 2). It indeed revealed 5 clusters with the largest cluster (1602 probes) comprising genes up-regulated by PCN in WT mice only (cluster 5). Another cluster (cluster 2) showed 407 probes down-regulated upon PCN treatment in WT mice only. Cluster 4 containing 498 probes show genes differentially regulated in WT vs Pxr-/- mice, independently of PCN. Finally, cluster 3 (605 probes) illustrates a PCN effect in both WT and Pxr-/- mice.”
5. In 2.3 section, how to activation PXR in Pparα (Pparαhep-/-) mice, is it treated with PCN in Pparαhep-/- mice? I'm confused about this part.
We are sorry if this part was confusing. We realize that the sentence that introduced this part of the results might have been ambiguous and we have therefore changed it (lines 174-176). The sentence “This prompted us to investigate whether PXR activation influenced PPARα activity” was changed to “This prompted us to investigate the intersection between PXR and PPARα activation to test the hypothesis that PXR activation influenced PPARα activity”.
6. FGF21 is a hepatokine – i.e., a hormone secreted by the liver. The author should detect FGF21 protein level in liver. And the author should consider that fasting time may influence FGF21 Level.
In mice, Fgf21 mRNA is expressed in many metabolic tissues including the liver but also, although to a much lower extent, in white and brown adipose tissues, skeletal muscle, pancreas and placenta (Kliewer SA, Mol Endocrinol, 2010). However it has been convincingly demonstrated that both in mice and humans circulating FGF21 mostly originates from the liver (Markan KR, Diabetes, 2014 ; Staiger H et al., 2017, Endocrine Reviews) and acts in an endocrine or autocrine/paracrine manner peripherally as well as centrally. Our lab has also provided evidence that PPARa deletion in hepatocytes was sufficient to alter both Fgf21mRNA levels in the liver and circulating FGF21 (Montagner A. et al. 2016 Gut, Iroz A. et al. 2017 Cell Reports) Therefore, we believe that the measurement of both hepatic Fgf21 mRNA levels and of the circulating protein is sufficient to demonstrate a significant impact of PXR activation on FGF21 regulation.
It is indeed well described that fasting time strongly influences FGF21 levels. At sacrifice, mice were killed in a randomized order, therefore, we did not expect significant difference in fasting state between our groups. However, to further confirm that the difference in circulating FGF21 levels observed in our study was not due to a difference in fasting duration between the different groups, we have now measured glucose levels in plasma and observed no significant difference between the different groups. These results were added in Supplementary Figure 1 and discussed in the results session as follow (lines 215-217): “These differences were not due to different fasting states since glycemia was not significantly different between the 4 groups (Supplementary Figure 1).”
Reviewer 2 Report
The study by Barretto et al. describes the gene expression inliver of mice treated with PXR activator pregnenolone 16α-carbonitrile (PCN). They obtained a microarray derived gene expression map of mice treated with PCN and, in order to study does changes specifically targeted by PXR activation, the authors compare gene expression in liver of PXR-/- mice treated in the same way. The authors find an interesting down regulation of PPARα-activated genes after PXR activation. Among this changes, FGF21 is reduced dramatically by PXR activation in serum.
The authors mention in the text that they have collected both liver and intestine samples. It would be interesting to know the transcriptional effects in intestine, as this tissue expresses PXR.
Could the authors comment on the effect of other lipid metabolic pathways, including cholesterol and fatty acid synthesis? It would be interesting that the authors comment on the observed differences in plasma lipid composition after PCN treatment. The authors should be able to relate this results with the observed gene expression map, or at least comment about this aspect in the text.
Do the authors have an explanation of the apparent contradictory effect of PXR activation and genetic ablation of PXR on FGF21. In both cases, FGF21 concentration in plasma or expression was decreased. (decreased level).
Apart from these main points, the authors should correct some minor mistakes in the text.
L161. Please, indicate the meaning of *** in graph 2b.
L162. “(c&e) Venn diagram”... Possibly (c&f)?
L170. Is Pparαhep-/- correct in this sentence? The wild type littermates are named Pparαhep+/+ in L184.
L210. Where the ND values below the limit of detection or rather not measured? If the first, I suggest to change the acronym from ND to UDL.
L427. Provide a full reference #7.
L446. Provide a full reference #15.
L327. Separate figures and units; 23 oC, not 23 oC.
Author Response
We would like to thank the reviewer for his/her insightful comments. We believe that his/her suggestions have increased the scientific quality and clarity of this manuscript.
Reviewer 2
16α-carbonitrile (PCN). They obtained a microarray derived gene expression map of mice treated with PCN and, in order to study does changes specifically targeted by PXR activation, the authors compare gene expression in liver of PXR-/- mice treated in the same way. The authors find an interesting down regulation of PPARα-activated genes after PXR activation. Among this changes, FGF21 is reduced dramatically by PXR activation in serum.
1. The authors mention in the text that they have collected both liver and intestine samples. It would be interesting to know the transcriptional effects in intestine, as this tissue expresses PXR.
We agree with the reviewer that investigating the transcriptional effects of PXR activation in the intestine is of great interest. However, this is beyond the scope of the present study that aimed to gain insights into the metabolic regulations of PXR activation, focusing on hepatic transcriptional regulations. We aim to perform another microarray experiment in the intestine samples that we have collected and to publish an additional paper about these intestinal regulations. We hope that the reviewer will agree with us that adding intestinal regulations in the present paper will confuse the main message of the study. We have changed the method section accordingly (lines 417-418): “At sacrifice, the liver and the 3 parts of the small intestine and the colon were removed” was changed to “At sacrifice, the liver was removed”.
2. Could the authors comment on the effect of other lipid metabolic pathways, including cholesterol and fatty acid synthesis? It would be interesting that the authors comment on the observed differences in plasma lipid composition after PCN treatment. The authors should be able to relate this results with the observed gene expression map, or at least comment about this aspect in the text.
We thank the reviewer for this comment. Regarding cholesterol, we have added a Supplementary Figure 4 that highlights our microarray results and qPCR confirmation on genes involved in cholesterol synthesis (Cyp7a1, Sqle and Hmgcr) and transport (Cd36, Abcg5, Abca1 and Ldlr). We also added a paragraph on the discussion section to comment on these results (lines 268-280): “First, PXR activation induced a very significant decrease in plasma cholesterol levels, and a significant increase in liver triglycerides and cholesterol esters (Figure 1). The pro-steatotic effects of acute PXR activation have been shown in many studies. However, its role in the regulation of cholesterol homeostasis is more controversial. The anti-HIV drug Efavirenz has been recently shown to induce steatosis and hypercholesterolemia, an effect that was absent in a model of hepatic deletion of PXR (12). These perturbations were thought to be mediated through increased fatty acid transport and cholesterol synthesis, via the PXR-dependent regulation of Cd36 and Sqle. In our data, we confirmed that PXR activation significantly affected Cd36 and other transporters involved in cholesterol transport (Supplementary Figure 4), but did not observe any regulation of genes involved in cholesterol biosynthesis such as Cyp7a1, Sqle and Hmgcr. This resulted in decreased circulating cholesterol. “
We have also an additional Supplementary Figure 3 that shows several genes involved in de novo lipogenesis.
3. Do the authors have an explanation of the apparent contradictory effect of PXR activation and genetic ablation of PXR on FGF21. In both cases, FGF21 concentration in plasma or expression was decreased. (decreased level).
We hypothesize that different mechanisms are involved when PXR is activated and when it is absent. Interestingly, we observe that this apparent contradictory effect of PXR activation and genetic ablation of PXR is not only limited to the regulation of FGF21, but also occurs on other PPARα target genes (Cyp4a10, Cyp4a14 and Vnn1, see Figure 4). These results therefore suggest that both PXR activation and silencing inhibit PPARα activity. Upon PCN treatment, it has been previously shown that the binding of PPARα to the loci of the Cyp4a genes was decreased (20), however the exact molecular mechanism is still unknown. Interestingly, the same contradictory effects were previously observed on de novo lipogenesis. In human HepG2 cells, PXR activation by rifampicin promoted steatosis via induction of SREBP-1 pathway (mainly SREBP-1a), whereas PXR silencing enhanced AKR1B10 expression, which subsequently stabilized the acetyl-CoA carboxylase, thereby promoting de novo lipogenesis (7). However, these mechanisms are probably species-specific as in our data, we did not observe this increase in AKR1B10 expression, whereas the SREBP-1 pathway was increased by PXR ablation and not by PCN treatment (see Supplementary Figure 3). Overall, this demonstrates that complex mechanisms occur on the regulation of lipogenic pathways by PXR activation and ablation and that this might also be true for the regulation of b-oxidation and PPARα activity.
The following sentences were added to the discussion section (lines 342-358):
“This apparent contradictory effect was not limited to the regulation of FGF21 but was also observed in other PPARα target genes (Figure 4), therefore, it seems that both PXR activation and silencing result in inhibition of PPARα activity, probably through distinct mechanisms that would need additional investigations. However, it is worth noticing that the same apparent contradictory effect was observed for the regulation of de novo lipogenesis. In human HepG2 cells, PXR activation by rifampicin promoted steatosis via induction of SREBP-1 pathway (mainly SREBP-1a), whereas PXR silencing enhanced AKR1B10 expression, which subsequently stabilized the acetyl-CoA carboxylase, thereby promoting de novo lipogenesis (7). However, these mechanisms are probably species-specific as in our data, we did not observe this increase in AKR1B10 expression, whereas the SREBP-1 pathway was increased by PXR ablation and not by PCN treatment (Supplementary Figure 3). Overall, this demonstrates that complex species-specific mechanisms occur in the regulation of lipogenic pathways by PXR activation and ablation and that this might also be true for the regulation of b-oxidation and PPARα activity.”
Apart from these main points, the authors should correct some minor mistakes in the text.
L161. Please, indicate the meaning of *** in graph 2b.
L162. “(c&e) Venn diagram”... Possibly (c&f)?
L170. Is Pparαhep-/- correct in this sentence? The wild type littermates are named Pparαhep+/+ in L184.
L210. Where the ND values below the limit of detection or rather not measured? If the first, I suggest to change the acronym from ND to UDL.
L427. Provide a full reference #7.
L446. Provide a full reference #15.
L327. Separate figures and units; 23 oC, not 23 oC.
We thank the reviewer and have corrected all those mistakes.
Reviewer 3 Report
Title: Gene expression profiling reveals that PXR activation inhibits hepatic PPARα activity and decreases FGF21 secretion in male C57Bl6/J mice
Authors studied the effect of PXR activation in the liver of wild-type vs Pxr-/- C57Bl6/J male mice treated with PCN. It was observed that PXR activation produced triglyceride deposition in the liver parenchyma and regulated the expression of genes associated with xenobiotic metabolizing enzymes. Importantly, they found that PPARα was downregulated. This effect may be associated role of PXR in energy homeostasis.
This is a well-performed study and conclusions are based on the obtained results. However, I have some concerns.
The main results from my point of view is that PXR is involved in steatohepatitis, which is a disease that is increasing worldwide and there if not an effective treatment to prevent/attenuate/reverse this disease. In this regard, several molecules/pathways are involved, SREBPs and PPAR-gamma are the most relevant. Therefore, authors are invited to study the relationship between PXR and these factors; perhaps some WB will add relevance to their study.
Perhaps, authors may discuss the importance of PXR in fatty liver induced by xenobiotics. Several drugs, toxins, and toxic plants may induce liver steatosis; therefore, it is possible that PXR is involved in such a process.
It would be interesting to see the livers and some histologies, like an H&E and ORO to evaluate the hepatic parenchyma and triglycerides accumulation.
Author Response
We would like to thank the reviewer for his/her insightful comments. We believe that his/her suggestions have increased the scientific quality and clarity of this manuscript.
Reviewer 3
Authors studied the effect of PXR activation in the liver of wild-type vs Pxr-/- C57Bl6/J male mice treated with PCN. It was observed that PXR activation produced triglyceride deposition in the liver parenchyma and regulated the expression of genes associated with xenobiotic metabolizing enzymes. Importantly, they found that PPARα was downregulated. This effect may be associated role of PXR in energy homeostasis.
This is a well-performed study and conclusions are based on the obtained results. However, I have some concerns.
The main results from my point of view is that PXR is involved in steatohepatitis, which is a disease that is increasing worldwide and there if not an effective treatment to prevent/attenuate/reverse this disease. In this regard, several molecules/pathways are involved, SREBPs and PPAR-gamma are the most relevant. Therefore, authors are invited to study the relationship between PXR and these factors; perhaps some WB will add relevance to their study.
We thank the reviewer for this comment. We have now added two Supplementary Figures with the microarray results and confirmatory qPCR experiments for genes involved in cholesterol transport and synthesis (Supplementary Figure 4) and for genes involved in de novo lipogenesis (Supplementary Figure 3), among which the lipogenic genes and Pparg. Upon PCN treatment, we did not observe any significant regulation in the expression of SREBPs not in their most classical target genes (Fas, Scd1…). Pparg was also not significantly changed upon PCN treatment or by PXR deletion. Therefore, given this lack of significant gene regulation and the fact that PXR is exclusively known as a transcriptional regulator, we did not consider appropriate to conduct additional western-blot experiments.
Perhaps, authors may discuss the importance of PXR in fatty liver induced by xenobiotics. Several drugs, toxins, and toxic plants may induce liver steatosis; therefore, it is possible that PXR is involved in such a process.
We have added the following sentences to the discussion sections (lines 366-370): “Indeed, PXR’s main target gene CYP3A4 is known to be involved in the metabolism of more than 60% of the currently marketed drugs (6) and several hundreds of environmental, occupational and natural products are demonstrated PXR agonists in both mice and humans (3). Therefore, regulation of hepatic lipid accumulation by acute or chronic PXR activation might be an important mechanism of xenobiotic-induced steatosis.”
It would be interesting to see the livers and some histologies, like an H&E and ORO to evaluate the hepatic parenchyma and triglycerides accumulation.
Unfortunately, we did not collect proper liver samples for histology and could only quantify steatosis based on biochemical assays such as triglyceride and cholesterol ester accumulation in the liver tissue.
Reviewer 4 Report
In the current study of Barretto et.al., the author compared the transcription profiles of PCN treated Pxr-/- and its control mice and found a panel of differentially expressed genes. The author further compared those target genes to a previously published microarray dataset containing Pparα dependent gene expression and made the assertion that PXR activation inhibits hepatic PPARα activity. Overall, the experiments performed are missing proper controls. The conclusions often are overstated. The aim of the study was distracted.
Major issues:
1. Line 74, Page 2. One of the aims of the study was set as to reveal the mechanisms of PXR-induced hepatic triglyceride accumulation. This need more focused study. So far the link between PXR-PPAR-FGF21 is unclear.
2. Line 76, Page 2, why only male Pxr-/- mice are used for comparison?
3. Line 78, Page2 and Figure 1. Though there were signs of lipid accumulation and liver inflammation, the author should provide HE staining of mice liver to show PCN treatment induced hepatic steatosis.
4. Figure 2, a. why in the PCA plot, the mice from same group form distinct clusters?
5. Figure 2, b. what is the connection of CYP expression to lipid accumulation?
6. Figure 3. The PXR and PPAR dependent transcriptome are not generated in parallel fashion. In PXR dataset, only male mice were used. In PPAR dataset, mice have been fasted for 24h. Bridging control experiments are needed to exclude those unequal treatment would not skew the data.
7. The resulted comparison of PXR and PPAR transcriptome was quite preliminary to prove that PPAR are inhibited by PXR. More functional study should be performed.
8. Figure 4, b. Can the author explain why activation or deletion of PXR could both lead to blockage of FGF21 expression? What is the consequence of modified FGF21 expression?
Round 2
Reviewer 1 Report
It's fine. The author answered my questions.
Reviewer 3 Report
Thank you for answering my questions.
Reviewer 4 Report
In the revised manuscript, Sharon et. al. have improved the paper by being more focused and more cautious in the overall merit of their data. Questions that were raised previously were either addressed experimentally or have been discussed. The overall quality of the paper have improved.